# Surgical Outcomes of Bilateral Inferior Rectus Muscle Recession for Restrictive Strabismus Secondary to Thyroid Eye Disease

**DOI:** 10.3390/jcm12216876

**Published:** 2023-10-31

**Authors:** Steffani Krista Someda, Naomi Umezawa, Aric Vaidya, Hirohiko Kakizaki, Yasuhiro Takahashi

**Affiliations:** 1Department of Oculoplastic, Orbital & Lacrimal Surgery, Aichi Medical University Hospital, Nagakute 480-1195, Aichi, Japan; steffsomeda@gmail.com (S.K.S.); nume0828@yahoo.co.jp (N.U.); aricvaidya1@gmail.com (A.V.); cosme_geka@yahoo.co.jp (H.K.); 2Department of Ophthalmology, Aichi Medical University, Nagakute 480-1195, Aichi, Japan; 3Department of Oculoplastic, Orbital & Lacrimal Surgery, Kirtipur Eye Hospital, Kathmandu 44600, Nepal

**Keywords:** bilateral inferior rectus muscle recession, nasal inferior rectus muscle transposition, restrictive strabismus, thyroid eye disease

## Abstract

This retrospective, observational study examined the surgical outcomes of bilateral inferior rectus (IR) recession in thyroid eye disease. Twelve patients who underwent bilateral IR muscle recession were included in the study. Surgical success was defined as patient achievement of the following conditions: (1) a postoperative angle of vertical ocular deviation of ≤3°; (2) a postoperative cyclotropic angle of ≤2°; (3) postoperative binocular single vision, including the primary position; and (4) postoperative enlargement of the field of binocular single vision. Linear regression analyses were performed to analyze the relationship between postoperative changes in the vertical and torsional ocular deviation angles and the amount of IR muscle recession and nasal transposition. Consequently, 9 out of 12 patients were deemed to have had successful surgical outcomes. There was a positive correlation between a change in the vertical deviation angle and a side-related difference in the amount of IR muscle recession in successful cases (crude coefficient, 2.524). A positive correlation was also found between a change in the torsional deviation angle and the amount of IR recession (crude coefficient, 1.059) and nasal transposition (crude coefficient, 5.907). The results will be helpful to more precisely determine the amount of recession and nasal transposition of the IR muscle in patients with thyroid-related bilateral IR myopathy.

## 1. Introduction

Thyroid eye disease (TED) is a debilitating condition, with approximately 15% of patients with TED suffering from symptomatic ocular motility disturbance due to fibrotic changes in the extraocular muscles [1]. Among the extraocular muscles, the inferior rectus (IR) and medial rectus (MR) are known to be commonly affected. When an active inflammation has subsided and the IR muscle becomes fibrotic, patients often complain of intractable diplopia, especially for vertical and torsional ocular misalignments. This leads to difficulty in performing daily activities, such as driving, reading, writing, and eating, thus affecting the quality of life in many patients. Patients elevate their chins and tilt their heads in order to compensate for vertical and torsional diplopia, while patients with severe vertical and torsional strabismus cannot obtain binocular single vision, despite compensatory head positioning or even spectacle correction using prism glasses [2]. The primary goal of strabismus surgery in patients with inactive TED is to achieve a substantial field of binocular single vision (BSV) in the primary, as well as in the downward, position [3,4,5]. However, the surgical outcomes of strabismus surgery in TED can be highly unpredictable, with 17% to 45% of cases requiring reoperation due to undercorrection, overcorrection, and postoperative torsional deviations [1,4,6]. One of the possible reasons for unpredictable surgical outcomes is the variation in the degree of fibrous changes and adipose degeneration in the extraocular muscles among patients with TED [2,7].

Bilateral IR myopathy is a relatively common condition associated with TED. In such cases, the bilateral restriction of supraduction and a large angle of excyclotropia make it more difficult to obtain BSV in any gaze direction [8]. Although IR muscle recession is usually performed to correct this condition, unilateral muscle surgery (only on the more severe eye) can cause progressive overcorrection due to increased stimulation in the contralateral superior rectus (SR) muscle, resulting in hypertropia of the ipsilateral eye, which can be explained by Hering’s law [6,8,9]. Sprunger and Helveston indicated that half of the patients with TED who underwent unilateral IR recession experienced progressive overcorrection [9]. Therefore, asymmetric bilateral IR muscle recession is a better option to overcome this problem. However, since the IR muscle is a secondary adductor [10], bilateral IR muscle recession occasionally leads to an A-pattern strabismus [11,12]. Dagi et al. determined that postoperative A-pattern strabismus was found when the IR was recessed more than 6 mm [4]. This is due to the recruitment of the superior oblique (SO) muscle during infraduction or the possibility of excessive SO tone. On the other hand, Jellema et al. claimed that their study revealed no real A-pattern postoperatively, but mentioned the presence of a reduced horizontal squint angle, especially in the downward (primary position, 1.0°; downward, 3.0°) [13].

In the past, there have been few reports regarding the surgical outcomes of bilateral IR recession in TED [1,6,8,13,14,15]. The surgical outcomes of bilateral IR muscle recession reported in these previous reports were favorable, with surgical success rates of 64–100%. However, most of these reports included either a small number of patients (3–8 patients) [1,6,8,14] or patients with a prior history of orbital decompression [1,13,15], which causes further restriction of extraocular muscle motility [16,17]. To date, only one published report investigated the factors affecting the surgical outcomes of bilateral IR muscle recession [13]. This report presented the dose–effect relationship between improved elevation and the amount of IR muscle recession. Although improvement in supraduction is a good outcome measurement, the deviation angle in the primary position would be a more important postoperative finding of strabismus surgery in TED because obtaining a substantial field of BSV in the primary position is indeed the primary goal in these cases. None of the previous studies determined the relationship between the changes in ocular deviation angles in the primary position and the amount of IR muscle recession and nasal transposition. Understanding this relationship will ensure a more tailored bilateral IR muscle recession for TED patients [2,7]. We have conducted this study to determine the effectiveness of asymmetric bilateral IR recession for the management of restrictive strabismus in patients diagnosed with TED who did not undergo orbital decompression.

## 2. Materials and Methods

### 2.1. Ethics Approval

The institutional review board (IRB) of Aichi Medical University Hospital approved this study, which was conducted in accordance with the tenets of the Declaration of Helsinki and its later amendments (approval number, 2023-022). The IRB granted a waiver of informed consent for this study, based on the ethical guidelines for medical and health research involving human subjects established by the Japanese Ministry of Education, Culture, Sports, Science, and Technology, and by the Ministry of Health, Labor, and Welfare. The waiver was granted because the study was a retrospective chart review, not an interventional study. Nevertheless, at the request of the IRB, an outline of the study, available for public viewing, was published on the Aichi Medical University website, which also gave the patients an option to refuse to participate in the study, although none of the patients did. Personal identifiers were removed from the records prior to data analysis.

### 2.2. Study Design

This retrospective, observational study included Japanese patients with TED who underwent asymmetric bilateral IR muscle recession, with or without IR muscle nasal transposition, at Aichi Medical University Hospital, Japan, performed by one of the authors (Y.T.) from January 2015 to February 2023. The restriction of upward gaze was graded on an ordinal scale (0 = duction > 45°, 1 = 30–45°, 2 = 15–30°, and 3 = <15°) [18,19], and patients with at least grade 1 in a less severe eye were included in this study. Bilateral positive forced duction tests were confirmed intraoperatively in all patients. Patients with a lack of clinical data, a history of strabismus or orbital decompression surgery, a follow-up period of less than 3 months, concomitant neuro-ophthalmologic disorder(s), and an intracranial lesion were excluded from this study. Since the outcome and its associated changes were best measured after a certain duration of follow-up from the time of surgical intervention, a retrospective, observational approach was deemed appropriate for this study.

### 2.3. Diagnosis of TED

A diagnosis of TED was based on the presence of at least one of the characteristic eyelid signs (eyelid fullness, eyelid retraction, and/or eyelid lag), as well as the presence of elevated thyroid antibody levels [10]. All patients included in this study were diagnosed with an autoimmune thyroid disorder by their endocrinologists prior to referral to our service. The IR muscle was confirmed to be enlarged, without muscle tendon involvement, on magnetic resonance images (MRI), and upward gaze was restricted on both sides in all patients, which also supported the diagnosis of TED. All patients included in the study were controlled as euthyroid at the time of surgery. We confirmed that all patients were in the static or chronic “burnout” phase of TED, based on the clinical activity score of 0 to 1 immediately before surgery and the absence of inflammation in the extraocular muscles on T2-weighted fat-suppressed MRIs obtained 3 months prior to the surgery.

### 2.4. Data Collection

The charts of all TED patients who underwent asymmetric bilateral IR recession surgery were reviewed from the electronic medical record (EMR) of the hospital. The following data from the EMR were collected: age, sex, smoking status, history of steroid pulse therapy and/or orbital radiotherapy, amounts of IR muscle recession and nasal IR muscle transposition, concomitant strabismus surgery, preoperative and postoperative angles of ocular deviation, and preoperative and postoperative fields of BSV. We asked all the patients the number of cigarettes smoked per day. Patients who previously smoked but stopped smoking cigarettes ≥2 years prior to the examinations were considered non-smokers [20]. The smoking status was classified by the number of cigarettes smoked per day, according to a report by Pfeilschifter and Ziegler, as follows: 0, no smoking; 1, ˂10 cigarettes/day; 2, 10–20 cigarettes/day; and 3, >20 cigarettes/day [21]. The dose used for orbital radiotherapy was 20 Gy in all the treated patients.

Preoperative and postoperative measurements of ocular deviation angles and the field of BSV were carried out by an exclusive orthoptist (N.U.) one day before and three months after surgery, respectively. The angle of ocular deviation was measured using a synoptophore (Clement Clarke International Ltd., Edinburgh, UK). The patient’s head was positioned upright, and the instrument was set such that the fixating eye was the eye on which surgery was planned. One of the two arms of the synoptophore was fixed at 0°. We used two slides: a black circle with a cross-shaped blank and a black cross (L-25G; Inami, Tokyo, Japan). The black circle slide was fixed in the arm of the fixating eye, and the patient was asked to move the black cross until it was positioned appropriately within the circle. We then recorded the angle at which this was achieved. The deviation angles were measured in the primary position, at the 15° upward gaze, and at the 15° downward gaze. The Goldmann perimeter (Haag Streit, Bern, Switzerland) was used to measure the field of BSV.

The areas of fields of BSV were measured using the freehand measuring tool available in ImageJ software ver. 1.49 (National Institute of Health, Bethesda, MD, USA). We first measured the normal area of BSV in Japanese, based on our previous study [22], and then determined the areas of pre- and postoperative fields of BSV (Figure 1). The percentages of pre- and postoperative fields of BSV were calculated against the normal field of BSV (%BSV). In addition, the results of the field of BSV were classified into five categories (B1 to B5), according to the methods used in our previous study [23], as follows: B1, within normal range (±2 × standard deviations); B2, the field of BSV reaches at least 20 degrees superiorly, 40 degrees inferiorly, and 30 degrees horizontally; B3, the field of BSV is smaller than that of B2, but includes primary gaze; B4, the field of BSV does not include primary gaze; B5, the field of BSV cannot be obtained in any direction of gaze.

### 2.5. MRI

MRI was performed using a 1.5-Tesla scanner (Magnetom Abant™; Siemens Healthcare, Erlangen, Germany), with the patients in the supine position. Coronal T1- and T2-weighted gradient-echo sequences were acquired (T1—repetition time: 500 ms, echo time: 10 ms, field of view: 140 × 140 mm, matrix: 256 × 220, section thickness: 3 mm with a 0.6 mm gap between slices; T2—repetition time: 4000 ms, echo time: 100 ms; all other parameters were the same as in T1). Patients were asked to look at a light source to ensure that their eyes were fixed in the primary position. The cross-sectional areas of the IR, SR, MR, and SO muscles on a coronal T1-weighted MRI image and those of the lateral rectus muscle on an axial T1-weighted MRI image at the largest point were measured by one of the authors (Y.T.), using the measuring tool available in the MRI viewer (ShadeQuest/ViewR™; Yokogawa Medical Solutions Corporation, Tokyo, Japan) (Figure 2) [24]. This study did not measure the cross-sectional area of the inferior oblique muscle because sagittal images could not be obtained from some of the patients [25].

### 2.6. Surgical Procedure

Surgery was performed under general anesthesia using the same method of IR muscle recession employed in our previous studies [2,7,10]. A perilimbal conjunctival incision, with radial relaxing incisions, was made in the inferior or inferonasal quadrant. A muscle hook was used to secure the IR muscle at its insertion, and the Tenon’s capsule around the IR muscle was thoroughly dissected using cotton swabs. The width of the IR muscle tendon was measured at the scleral insertion using a caliper. The IR muscle tendon was secured using locking 6-0 or 8-0 polyglactin sutures (Vicryl^®^; Johnson and Johnson Company, New Brunswick, NJ, USA) at two points 1 mm posterior to the globe insertion because of the 1 mm tip thickness of the muscle hook. Then, the IR muscle was detached from its insertion. The sutures were fixed to the sclera 1 mm posterior to the point that was estimated based on the preoperative angle of the vertical ocular deviation and the grade of upward gaze restriction. More hypotropic eyes with a more severe restriction of supraduction were defined as more severe eyes. The maximum amount of IR recession in a more severe eye was set at 8 mm [26,27]. The amount of IR recession in a less severe eye was commonly set at 3 mm [1]. In patients with grade 1 in a less severe eye, it was set at 2 mm. The recession of the IR muscle in a more severe eye was calculated, based on the following formula: 2° of hypotropic angle per 1 mm IR muscle recession + the amount of IR recession in a less severe eye [7]. When patients were aware of torsional diplopia before surgery and when the preoperative excyclotropia angle was larger than the estimated excyclotropic angle correction after the inferior rectus muscle, calculated with the undermentioned formula, we transposed the IR muscle nasally along the spiral of Tillaux. The amount of nasal IR muscle transposition was preoperatively calculated based on the preoperative angle of excyclotropia and the measurement result of the tendon width as follows: 8° of excyclotropic angle per one IR muscle tendon width transposition +0.4° of excyclotropic angle per 1 mm IR muscle recession [2]. In anticipation of late overcorrection, we set the target angles of the vertical and torsional deviations with the remaining 2–3° of hypotropia and 1–2° of excyclotropia. The IR muscle tendon was additionally fixed to the sclera using 6-0 or 8-0 polyglactin sutures at two to four points to prevent the slippage of the muscle. Finally, the conjunctiva was closed using 8-0 polyglactin sutures.

### 2.7. Statistical Analyses

Patient data and measurement results were expressed as the means ± standard deviations. To evaluate the pattern strabismus, the horizontal deviation angle measured in a 15° downward gaze was subtracted from that measured in a 15° upward gaze. The values were expressed as positive and negative for the tendency of A- and V-pattern strabismus, respectively. As we roughly set the measurement of two prism diopters corresponding to 1°, subtraction values of 5° and −7.5° or greater were considered to indicate clinically significant A- and V-pattern strabismus, respectively [28]. Surgical success was defined as patient achievement of the following four conditions: (1) a postoperative angle of vertical ocular deviation of ≤3°; (2) a postoperative cyclotropic angle of ≤2°; (3) a postoperative BSV grade of B3 or better; and (4) postoperative improvement of %BSV. Comparisons of patient age, ratio of patients with history of steroid pulse therapy/orbital radiotherapy, ratio of smoking status, amount of IR muscle recession and nasal transposition, ratio of patients who underwent additional strabismus surgery, cross-sectional areas of the extraocular muscles in more and less severe eyes, preoperative ocular deviation angles, preoperative pattern strabismus, and preoperative BSV grade between the successful and unsuccessful groups were conducted via the Mann–Whitney U test, Fisher’s exact test, or the Chi-square test. Due to the small sample size, the Mann–Whitney U test and Fisher’s exact test were used for the comparison of independent samples and the analyses of 2 × 2 tables, respectively, rather than the Student’s *t*-test or the Chi-square test. Univariate linear regression analysis was performed to analyze the relationship between the side-related difference in the amount of IR muscle recession and the postoperative changes in the vertical ocular deviation angle. Univariate and subsequent multivariate linear regression analyses, with stepwise variable selection, were performed to identify factors influencing changes in torsional deviation angles. The predictive variables investigated included the amount of IR muscle nasal transposition in both more and less severe eyes and the sum of the amount of IR muscle recession in both eyes. We conducted linear regression analyses in all cases. All statistical analyses were performed using SPSS™ version 26 software (IBM Japan, Tokyo, Japan). Two-tailed *p* values < 0.05 were deemed to indicate statistical significance.

## 3. Results

Data regarding patient characteristics, surgery, and measurements are shown in Table 1. This study included 12 patients (2 males and 10 females; patient age, 60.6 ± 8.6 years). Six patients exhibited more severe right eyes, while six patients possessed more severe left eyes.

The amount of IR recession was 6.1 ± 1.7 mm in the more severe eyes. In the less severe eyes, the amount of IR recession was set at 2 mm in 2 patients and at 3 mm in 10 patients. The side-related difference in the amount of IR recession was 3.3 ± 1.5 mm. The amount of IR muscle nasal transposition was 0.2 ± 0.3 muscle width in more severe eyes and 0.1 ± 0.1 muscle width in less severe eyes. Bilateral and unilateral (more severe eyes) IR muscle nasal transpositions were performed in 2 and 3 patients, respectively, while the other 7 patients did not undergo nasal transposition of the IR muscle. Bilateral and unilateral MR muscle recession was performed in 1 and 2 patients, respectively, for correction of concomitant esotropia.

The surgical outcomes are shown in Table 2. Before surgery, the angles of hypotropia, esotropia, excyclotropia, and pattern deviation were 6.2 ± 3.8°, 6.0 ± 5.3°, 5.4 ± 3.7°, and 0.8 ± 3.8°, respectively. A-pattern strabismus was present in 3 patients. Preoperative %BSV was 19.6 ± 21.4%, and preoperative BSV grade was B3 in 4 patients, B4 in 4 patients, and B5 in 4 patients, respectively. After surgery, the angles of hypotropia, esotropia, excyclotropia, and pattern deviation decreased to 2.3 ± 3.6°, 1.0 ± 3.2°, −0.1 ± 1.4°, and −0.5 ± 3.0°, respectively. A-pattern strabismus was corrected after bilateral IR muscle recession, without nasal transposition, in the 3 patients due to a greater reduction of the horizontal deviation angle in the upward gaze compared to that in downward gaze. None of the patients with A-pattern strabismus exhibited incyclotropia before or after surgery. None of the other 9 patients developed new-onset pattern strabismus. Postoperatively, the %BSV increased to 45.5 ± 26.3%, and the BSV grade was B1 in 1 patient, B2 in 5 patients, B3 in 3 patients, and B4 in 3 patients.

A comparison of successful and unsuccessful cases is shown in Table 1 and Table 3. Nine patients (75.0%) were deemed as successful surgical cases. The other 3 patients were considered unsuccessful cases due to the undercorrection of hypotropia. In 2 of these 3 cases, excyclotropia was adequately corrected, but the other 1 case showed the overcorrection of excyclotropia (−4 degrees). Although none of the measurements were significantly different based on statistical analysis, the amount of IR recession in more severe eyes seemed to be larger in unsuccessful cases (5.7 ± 1.8 mm vs. 7.3 ± 1.2 mm; *p* = 0.209). Additional SR muscle surgery was performed in the 3 unsuccessful cases, after which all the 3 cases obtained a field of BSV of B3 or better.

The results of the linear regression analyses are shown in Table 4. Univariate analysis for the correction of hypotropia showed that, although the change in the vertical ocular deviation angle was not correlated with the side-related difference in the total amount of IR muscle recession (*p* = 0.125), there was a correlation between the side-related difference and the amount of IR muscle recession in successful cases (*p* < 0.001) (adjusted R^2^ = 0.923; *p* < 0.001). The crude coefficient of the side-related difference in the amount of IR muscle recession was 2.524. Multivariate stepwise analysis showed that the change in the cyclotropic angle was correlated with the amount of IR nasal transposition in more severe eyes (*p* = 0.012) and the sum of the total amount of IR muscle recession (*p* = 0.005) (adjusted R^2^ = 0.817; *p* < 0.001). The crude coefficients of the amount of IR muscle nasal transposition in more severe eyes and the sum of the amount of IR muscle recession were 5.907 and 1.059, respectively. Multivariate stepwise analysis also showed that a change in the cyclotropic angle was correlated with the sum of the amount of IR muscle recession (*p* = 0.006) in successful cases (adjusted R^2^ = 0.689; *p* = 0.006). The crude coefficient of the sum of the amount of IR muscle recession was 1.000.

## 4. Discussion

This study investigated the surgical outcomes of bilateral asymmetric IR recession in TED and the factors influencing them. Surgical success was obtained in 75% of patients in our study. Cruz and Davitt reported that six out of eight patients were found to have successful correction of hypotropia, while the other two patients remained undercorrected [6]. Flanders and Hastings presented their surgical outcomes, with all three patients obtaining complete correction of hypotropia, but one of the three patients underwent symmetric IR recession [14]. Another patient underwent a second surgery, but the amount of IR recession was not mentioned [14]. Arnolds and Reynolds reported their surgical outcomes, with all four patients deemed as surgical success, but the detailed surgical methods were not presented [8]. In the report by Jellema et al., 64% of patients required no further surgery after bilateral IR muscle recession, while 17% of patients needed one or more additional vertical strabismus surgeries [13]. Volpe et al. demonstrated that 45 out of 54 patients (84%) obtained a final vertical deviation angle of <5 prism diopters, but they combined patients who underwent unilateral and bilateral IR muscle recession ± contralateral SR muscle recession [1]. The surgical outcomes of our study were comparable to those of the aforementioned studies.

Previous studies have reported that an enlarged SR muscle can cause failure of bilateral IR muscle recession due to underestimation of excyclotorsion and late overcorrection of hypotropia [4,5,6,8]. Although there were three unsuccessful cases in this study, those three patients showed that the undercorrection of hypotropia and the cross-sectional areas of the SR muscle in both eyes were not significantly different between the successful and unsuccessful cases. The difference may not have been statistically significant, but the amount of IR recession in the more severe eyes seemed to be larger in the unsuccessful cases (5.7 ± 1.8 mm vs. 7.3 ± 1.2 mm). A large amount of IR recession in the more severe eye may cause divergence between the expected and actual dose–effect response of bilateral IR muscle recession.

The change in the vertical ocular deviation angle was correlated with the side-related difference in the amount of IR muscle recession in the successful cases, and the crude coefficient of the side-related difference was 2.524. This means that for every 1 mm of difference in the amount of IR recession between the right and left eyes, there are 2.524 degrees of hypotropia improvement in the more severe eye. This coefficient was close to the one reported in our previous study, showing the mean dose–effect relationship of unilateral IR muscle recession to be 2.27 ± 0.6 degrees/mm [7]. Jellema et al. further reported on the surgical outcomes of bilateral IR muscle recession, showing the dose–effect response to be 1.7 ± 1.7° for every millimeter of recession [13]. However, since this dose–effect response was only noted for improved elevation, it was not comparable with the coefficient shown in our study. Moreover, Jellema’s report included patients with a prior history of orbital decompression.

The change in torsional angle was correlated with the amount of IR nasal transposition in the more severe eyes (crude coefficient of 5.907), which was notably smaller compared to that in our previous study showing the results of unilateral IR muscle recession and nasal transposition in TED (crude coefficient of 8.546) [2]. On the other hand, the correlation of the torsional angle change and the sum of the amount of IR muscle recession (crude coefficient of 1.059) seemed to be larger than the coefficient derived from unilateral IR muscle recession (crude coefficient of 0.405) reported in our previous study [2]. In this present study, the amount of IR muscle nasal transposition was preoperatively calculated based on the result of unilateral IR muscle recession shown in our previous study, which was 8° of excyclotropic angle per one IR muscle tendon width transposition + 0.4° of excyclotropic angle per millimeter of IR muscle recession [2]. However, the results of unilateral IR muscle recession and nasal transposition cannot be applied to patients who require bilateral IR muscle recession. The study by Jellema et al. also showed improvement in the squint angle with regards to primary gaze, as well as reduced excyclotorsion, with a dose–effect response of 0.74 ± 0.61° per millimeter of recession in the primary position, postoperatively [13]. This value was relatively intermediate between the values shown in the present study and those in our previous study [2].

Three patients exhibited A-pattern strabismus before surgery, which was corrected after bilateral IR recession, without IR nasal transposition. In addition, none of the other nine patients experienced new-onset pattern strabismus after surgery. Kushner, in his study involving 51 patients with symptomatic diplopia on downward gaze at near vision, claimed that bilateral IR muscle recession in TED poses the risk of A-pattern strabismus and nasal transposition of the recessed IR muscle, as the prevention of A-pattern exotropia would aggravate incyclotropia [11]. However, since our previous study indicated that unilateral IR muscle recession ± IR nasal transposition does not increase the risk of pattern strabismus [10], the same conclusion may be applicable to bilateral IR muscle recession. There were no sources explaining the resolution of A-pattern strabismus after IR recession without nasal transposition. Theoretically, however, the restriction of the fibrotic IR muscle could have been causing compensatory overaction of the antagonist SR muscle, resulting in incyclotorsion and/or adduction on elevation, and the weakening of the IR muscle eventually led to the resolution of the SR overaction as well. On the other hand, a previous study showed that a thicker IR muscle caused A-pattern strabismus after IR muscle recession in TED [15]. We compared the thickness of the IR muscle between cases with and without A-pattern strabismus. It was noted that the IR muscle in more severe eyes was thinner in cases with A-pattern strabismus (*p* = 0.036), although the thickness was not different between cases with and without A-pattern strabismus in less severe eyes (*p* = 0.864). This may also affect the improvement of A-pattern strabismus in our study. Furthermore, nasal transposition of the IR muscle has been proven beneficial, both to correct excyclotropia and to prevent A-pattern strabismus [13], and setting the target torsional angle to 1–2° of undercorrection may prevent aggravating the incyclotropia after surgery.

This study has some limitations, one of which is a relatively small sample size, with only 12 patients included in the analysis. A small sample size can limit the statistical power of the study and may not adequately represent the diverse characteristics and variations in TED. Another limitation of this study is the short follow-up period. Postoperative ocular deviation angles and BSV results were obtained after only 3 months. The authors recommend a larger sample size and a longer duration of follow-up to gather more data on the surgical outcomes, as well as to account for the presence of late overcorrection. A retrospective study with a 2-year follow-up period was found to confirm the presence of late overcorrection in TED patients who underwent unilateral IR muscle recession for restrictive strabismus [5]. Previous studies have shown that IR muscle recession can result to lower eyelid retraction [29,30]. There was no data regarding changes in the lower eyelid position after surgery, hence presenting another limitation of the study. Furthermore, the authors used an ordinal scale of 0 to 3 as the grading system for ocular motility restriction, instead of the usual 0 to 4. A more universal grading system should probably be adopted for consistency across all published literature.

## 5. Conclusions

This study investigated the surgical outcomes of asymmetric bilateral IR muscle recession in TED. A total of 9 out of 12 patients (75.0%) were judged as successful surgical cases and obtained the field of binocular single vision in the primary position. In contrast, a large amount of IR muscle recession in a more severe eye might be the cause of undercorrection of hypotropia in the three unsuccessful cases. The coefficients of the amount of IR muscle recession and IR nasal transposition were shown for the correction of hypotropia and excyclotropia. This study indicates that bilateral IR muscle recession ± nasal transposition of the IR muscle proved to be beneficial in TED patients with vertical and torsional ocular deviation caused by fibrosis of the IR muscle, which can lead to debilitating diplopia. The results of linear regression analyses in this study will be helpful to more precisely determine the amount of IR muscle recession and IR nasal transposition required in TED patient with bilateral IR myopathy suffering from debilitating diplopia and to prevent the occurrence of postoperative ocular misalignments.

## Figures and Tables

**Figure 1 jcm-12-06876-f001:**
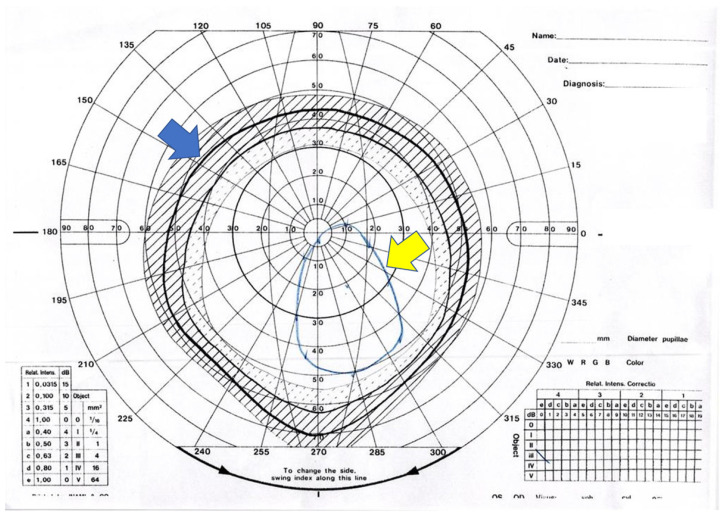
Measurements of normal (blue arrow) and actual (yellow arrow) fields of binocular single vision (BSV). Dashed shaded area means the range of ±1 standard deviation value of the measurement results of the field of BSV in normal Japanese volunteers [22]. Dotted area means there is a range of −2 standard deviation value.

**Figure 2 jcm-12-06876-f002:**
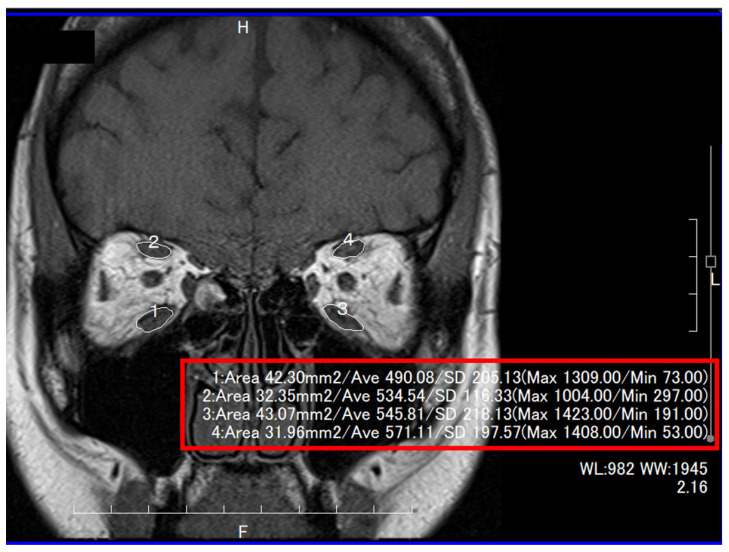
Measurements of the cross-sectional areas of inferior (#1 and #3) and superior (#2 and #4) recti muscles.

**Table 1 jcm-12-06876-t001:** Data for patient characteristics, surgery, and measurement results.

	Total	Successful	Unsuccessful	*p* Value
Patient number	12	9	3	
M/F	2/10	2/7	0/3	1.000
Patient age (years) (range)	60.6 ± 8.6 (45–75)	59.6 ± 9.4 (45–75)	63.7 ± 5.7 (59–70)	0.373
History of steroid pulse/orbital radiation therapies (Y/N)	8/4	7/2	1/2	0.236
Smoking status				
Non-smoker	9	7	2	0.157
<10 cigarettes/day	0	0	0	
11–20 cigarettes/day	2	2	0	
>20 cigarettes/day	1	0	1	
More severe eyes (R/L)	6/6	5/5	1/2	
Amount of IR muscle recession in more severe eyes (mm) (range)	6.1 ± 1.7 (3.0–8.0)	5.7 ± 1.8 (3.0–8.0)	7.3 ± 1.2 (6.0–8.0)	0.209
Amount of IR muscle recession in less severe eyes				
2 mm	2	2	0	1.000
3 mm	10	7	3	
Side-related difference in amount of IR muscle recession (mm) (range)	3.3 ± 1.5 (1.0–5.0)	2.9 ± 1.5 (1.0–5.0)	4.3 ± 1.2 (3.0–5.0)	0.209
Amount of IR muscle nasal transposition (muscle width) (range)				
More severe eyes	0.2 ± 0.3 (0–1.0)	0.2 ± 0.3 (0–0.5)	0.3 ± 0.6 (0–1.0)	1.000
Less severe eyes	0.1 ± 0.1 (0–0.3)	0.1 ± 0.1 (0–0.3)	0.1 ± 0.2 (0–0.3)	0.600
Additional treatment				
Unilateral MR muscle recession	2	1	1	0.595
Bilateral MR muscle recession	1	1	0	
Cross-sectional area of IR muscle (mm^2^) (range)				
More severe eyes	64.8 ± 15.7 (42.3–101.3)	61.5 ± 12.2 (42.3–77.3)	74.5 ± 23.7 (56.3–101.3)	0.727
Less severe eyes	61.1 ± 15.6 (41.3–86.5)	61.6 ± 15.9 (41.3–86.5)	59.8 ± 17.7 (43.5–78.6)	1.000
Cross-sectional area of SR muscle (mm^2^) (range)				
More severe eyes	27.4 ± 7.5 (10.0–36.2)	28.1 ± 5.3 (19.6–35.6)	25.3 ± 13.7 (10.0–36.2)	0.864
Less severe eyes	31.6 ± 10.0 (12.6–47.7)	32.6 ± 9.0 (20.7–47.7)	28.4 ± 14.3 (12.6–40.2)	1.000
Cross-sectional area of MR muscle (mm^2^) (range)				
More severe eyes	41.1 ± 8.6 (30.5–60.8)	39.6 ± 5.9 (33.9–49.8)	45.7 ± 15.1 (30.5–60.8)	0.727
Less severe eyes	42.2 ± 7.7 (34.0–56.3)	41.7 ± 7.1 (34.0–54.6)	43.8 ± 10.8 (36.6–56.3)	0.727
Cross-sectional area of SO muscle (mm^2^) (range)				
More severe eyes	17.0 ± 6.9 (8.4–30.9)	17.2 ± 5.0 (12.3–26.3)	16.3 ± 12.7 (8.4–30.9)	0.482
Less severe eyes	17.4 ± 4.2 (12.0–26.8)	17.0 ± 4.5 (12.0–26.8)	18.7 ± 3.9 (14.5–22.3)	0.482
Cross-sectional area of LR muscle (mm^2^) (range)				
More severe eyes	130.0 ± 29.8 (80.0–165.8)	132.4 ± 28.5 (80.0–165.8)	122.5 ± 39.3 (82.6–161.0)	0.864
Less severe eyes	127.6 ± 31.7 (84.8–167.3)	128.6 ± 31.0 (84.8–164.8)	125.5 ± 40.9 (85.6–167.3)	1.000

M, male; F, female; Y, yes; N, no; R, right; L, left; IR, inferior rectus; MR, medial rectus; SR, superior rectus; MR, medial rectus; SO, superior oblique; LR, lateral rectus.

**Table 2 jcm-12-06876-t002:** Surgical outcomes.

	Preoperative	Postoperative
Ocular deviation angle (degrees) (range)		
Hypotropia	6.2 ± 3.8 (1–13)	2.3 ± 3.6 (−1–10)
Esotropia	6.0 ± 5.3 (0–15)	1.0 ± 3.2 (−7–6)
Excyclotropia	5.4 ± 3.7 (1–15)	−0.1 ± 1.4 (−4–2)
Magnitude of pattern strabismus	0.8 ± 3.8 (−5–6)	−0.5 ± 3.0 (−7–4)
Pattern strabismus		
A-pattern	3	0
V-pattern	0	0
%BSV (%)	19.6 ± 21.4 (0–56.4)	45.5 ± 26.3 (9.0–86.0)
Grade of BSV		
B1	0	1
B2	0	5
B3	4	3
B4	4	3
B5	4	0

BSV, binocular single vision.

**Table 3 jcm-12-06876-t003:** Comparison of surgical outcomes between successful and unsuccessful cases.

	Successful (*n* = 9)		Unsuccessful (*n* = 3)			*p* Value: vs. Preoperative Values
	Preoperative	Postoperative	Preoperative	Postoperative	After Additional Surgery	
Ocular deviation angle (degrees) (range)						
Hypotropia	5.8 ± 3.9 (0–13)	0.3 ± 0.7 (−1–1)	7.3 ± 4.0 (3–11)	8.0 ± 1.7 (7–10)	2.0 ± 1.7 (1–4)	0.727
Esotropia	6.0 ± 5.5 (0–15)	1.8 ± 2.2 (−1–6)	6.0 ± 5.6 (1–12)	−1.3 ± 4.9 (−7–2)	1.0 ± 2.6 (−2–3)	0.864
Excyclotropia	4.4 ± 2.4 (1–9)	0.2 ± 0.7 (0–2)	8.3 ± 5.8 (5–15)	−1.0 ± 2.6 (−4–1)	0 ± 1.7 (−2–1)	0.282
Magnitude of pattern strabismus	0.8 ± 4.0 (−5–6)	−0.9 ± 3.0 (−7–2)	0.7 ± 3.8 (−2–5)	0.7 ± 3.5 (−3–4)		1.000
Pattern strabismus						
A-pattern	2	0	1	0		0.491
V-pattern	0	0	0	0		
%BSV (%)	20.9 ± 24.1 (0–56.4)	55.7 ± 21.5 (25.3–86.0)	15.9 ± 12.9 (1.0–24.1)	14.8 ± 7.9 (9.0–23.8)	43.5 ± 16.5 (31.2–62.2)	0.727
Grade of BSV						
B1	0	1	0	0	0	0.449
B2	0	5	0	0	1	
B3	3	3	1	0	2	
B4	2	0	2	3	0	
B5	4	0	0	0	0	

BSV, binocular single vision.

**Table 4 jcm-12-06876-t004:** Results of univariate and multivariate analyses.

Changes in Vertical Ocular Deviation Angle	Univariate					
Total	*p* Value	Crude Coefficient	95% CI			
Side-related difference in amount of IR muscle recession	0.125	1.548	−0.513 to 3.610			
Successful cases						
Side-related difference in amount of IR muscle recession	<0.001	2.524	1.874 to 3.173			
Changes in torsional angle	Univariate			Multivariate		
Total	*p* value	Crude coefficient	95% CI	*p* value	Crude coefficient	95% CI
Amount of nasal transposition of IR muscle in more severe eyes	0.007	8.788	3.017 to 14.559	0.012	5.907	1.661 to 10.153
Amount of nasal transposition of IR muscle in less severe eyes	0.004	21.818	8.525 to 35.111			
Sum of amount of IR muscle recession	0.002	1.427	0.634 to 2.220	0.005	1.059	0.417 to 1.702
Successful cases						
Amount of nasal transposition of IR muscle in more severe eyes	0.072	8.069	−0.936 to 17.074			
Amount of nasal transposition of IR muscle in less severe eyes	0.034	16.288	1.676 to 30.900			
Sum of amount of IR muscle recession	0.006	1.000	0.339 to 1.601	0.006	1.000	0.339 to 1.601

IR, inferior rectus; CI, confidence interval.

## Data Availability

Data supporting the results of this study are available on request.

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
