# Peer review of "Surgical Outcomes of Bilateral Inferior Rectus Muscle Recession for Restrictive Strabismus Secondary to Thyroid Eye Disease"

_jcm, 2023, doi:10.3390/jcm12216876_

Round 1

Reviewer 1 Report (New Reviewer)

Comments and Suggestions for Authors

I congratulate the authors to this very interesting paper. Please find my comments in the document.

Author Response

Reviewer #1

I congratulate the authors to this very interesting paper. Please find my comments in the document.

  1. Line 23. You should add the results of the dose effect analysis: amount of deviation reduction per mm recession as well as for nasal transposition. It’s very interesting.

Reply: Thank you for your suggestion. We added each crude coefficient in the abstract (page 1, lines 23-25).

  1. Line 45. MR should also be mentioned.

Reply: Thank you for your suggestion. We added “medial rectus” in page 1, line 35.

  1. Line 57. Might consider to add PMID: 29557700.

Reply: Thank you for your suggestion. This paper showed the limited dose of inferior rectus muscle recession. We, therefore, cited this paper in page 5, lines 195-196, as reference #26.

  1. Line 134. What about the MR and SR? Also enlarged?

Reply:  Since it is difficult to judge enlargement of the superior oblique muscle, unlike the rectus muscles, we added the measurement results of the cross-sectional areas of the medial rectus, superior oblique, and lateral rectus muscles in page 4, lines 172-178, and Table 1. We did not measure that of the inferior oblique muscle because sagittal images could not be obtained from some of the patients.

  1. Line 137. Clinically also inactive (swelling, chemosis, etc.)? For how long before surgery?

Reply: Clinical activity score was 0-1 immediately prior to the surgery in all patients. MRI was taken 3 months before the surgery. We added this information in page 3, lines 122-124.

  1. Line 154. Great!

Reply: Thank you for your comment.

  1. Line 165. Very good deviation and BSV measurement. Only thing I am missing is the motility measurement. The graded approach 1-3 seems a bit rough. For future studies you might consider the Kestenbaum glasses or Goldmann perimeter.

Reply: Thank you for your suggestion. As the reviewer indicated, ordinary scale of extraocular motility we used was rough. We used Goldmann perimeter to measure the field of BSV. We added the following comment in page 3, lines 149-150: The Goldmann perimeter (Haag Streit, Bern, Switzerland) was used to measure the field of BSV.

  1. Line 250. Please also add range here or in the table.

Reply: Thank you for your suggestion. We added the ranges of patient age, the amount of recession/transposition, and measurement results in the tables.

  1. Line 270. For these add also the range in the table.

Reply: Thank you for your suggestion. As the reply to comment #8, we added the ranges of patient age, the amount of recession/transposition, and measurement results in the tables.

  1. Line 290. Did they have more enlarged SR?

Reply: As shown in Table 1, the cross-sectional areas of the SR on both sides were not different between successful and unsuccessful cases.

  1. Lines 324-325. Move to the beginning of this paragraph.

Reply: Thank you for your suggestion. But the following phrase “which was comparable to the aforementioned studies” cannot be presented before presentation of the surgical outcomes in previous studies. We, therefore, only move the following sentence “Surgical success was obtained in 75% of patients in our study” to page 9, lines 309-310.

  1. Line 330. Interesting. Perfect. Forgot the comment in the results regarding this.

Reply: Thank you for your comment. We have also stated our response to comment #10.

  1. Line 332. Esser, Schittkowski et al. (2011) postulated that more than 9mm of IR recession do not lead to a higher effect and such muscles should elongated with bovine pericardium (Tutopatch).

Reply: Thank you for your comment. To reply to comment #3 and #8, we showed the range of the amount of IR recession in more severe eyes in Table 1. We set the maximum amount of IR recession at 8 mm, as the reviewer indicated. We added the following sentence “The maximum amount of IR recession in a more severe eye was set at 8 mm” in page 5, lines 195-196, and cited the above-mentioned paper as reference #27.

Reviewer 2 Report (New Reviewer)

Comments and Suggestions for Authors

This retrospective observational study offers valuable insights into the surgical management of thyroid eye disease-related vertical strabismus through bilateral inferior rectus (IR) muscle recession. The study reports that 75% of the patients achieved successful surgical outcomes, which included obtaining the field of binocular single vision in the primary position. The study also highlights the potential factors influencing surgical outcomes, such as the amount of IR muscle recession in the more severe eye.

While the study on surgical outcomes of asymmetric bilateral IR muscle recession in thyroid eye disease (TED) presents valuable insights, there are some weaknesses that should be considered:

1.     Small Sample Size: The study includes only 12 patients, which is a relatively small sample size. Such a limited sample size can limit the statistical power of the study and may not adequately represent the diverse characteristics and variations in TED cases. A larger sample size would provide more robust and generalizable results.

2.     Short Follow-Up Period: The study mentions that postoperative evaluations were conducted only three months after surgery. A longer follow-up period is necessary to assess the long-term stability of surgical outcomes. TED is a chronic condition, and its effects may evolve over time, so a more extended follow-up would provide a more comprehensive understanding of the outcomes.

3.     Limited Outcome Measures: The study primarily focuses on correcting hypotropia and excyclotropia. However, TED can have various other clinical manifestations, such as proptosis, eyelid retraction, and diplopia in different gazes. The study's narrow focus on specific aspects of strabismus may not capture the full spectrum of TED-related ocular issues.

4.     Ordinal Scale for Grading: The study mentions that the study used an ordinal scale of 0 to 3 for grading ocular motility restriction, deviating from the more commonly used 0 to 4 scale. This deviation from the standard scale could potentially introduce confusion and limit the comparability of the study's results with other research in the field. A justification for this choice is needed.

5.     Limited Generalizability: TED can manifest differently in patients, and factors such as disease severity and the presence of other associated eye conditions can vary widely. The study's findings may not be directly applicable to all TED patients, as the sample may not represent the full diversity of TED cases seen in clinical practice.

In summary, while the study provides valuable insights, these weaknesses, such as the small sample size, short follow-up period, and lack of detailed surgical methods, should be considered when interpreting the results and designing future research. Addressing these limitations could enhance the study's validity and applicability to a broader range of TED patients.

Comments on the Quality of English Language

Minor editing of English language required.

Author Response

Reviewer #2

This retrospective observational study offers valuable insights into the surgical management of thyroid eye disease-related vertical strabismus through bilateral inferior rectus (IR) muscle recession. The study reports that 75% of the patients achieved successful surgical outcomes, which included obtaining the field of binocular single vision in the primary position. The study also highlights the potential factors influencing surgical outcomes, such as the amount of IR muscle recession in the more severe eye.

While the study on surgical outcomes of asymmetric bilateral IR muscle recession in thyroid eye disease (TED) presents valuable insights, there are some weaknesses that should be considered:

  1. Small Sample Size: The study includes only 12 patients, which is a relatively small sample size. Such a limited sample size can limit the statistical power of the study and may not adequately represent the diverse characteristics and variations in TED cases. A larger sample size would provide more robust and generalizable results.

Reply: Thank you for pointing this out. Although the present study included a relatively larger number of patients, compared to previous studies involving only 3 to 8 cases, the number of patients was still small, as the reviewer has mentioned. We added the following sentence in the limitation section in page 11, lines 389-391: A small sample size can limit the statistical power of the study and may not adequately represent the diverse characteristics and variations in TED.

  1. Short Follow-Up Period: The study mentions that postoperative evaluations were conducted only three months after surgery. A longer follow-up period is necessary to assess the long-term stability of surgical outcomes. TED is a chronic condition, and its effects may evolve over time, so a more extended follow-up would provide a more comprehensive understanding of the outcomes.

Reply: Thank you for pointing this out. The authors have also acknowledged such limitation, as mentioned in page 11, lines 391-397. We are hoping that this study will be a stepping stone for future investigations to explore the long-term outcomes of bilateral IR recessions in TED.

  1. Limited Outcome Measures: The study primarily focuses on correcting hypotropia and excyclotropia. However, TED can have various other clinical manifestations, such as proptosis, eyelid retraction, and diplopia in different gazes. The study's narrow focus on specific aspects of strabismus may not capture the full spectrum of TED-related ocular issues. 

Reply: Thank you for your valuable comment. Indeed, TED presents with various clinical manifestations as the reviewer has mentioned. However, the focus and aim of this study was to examine the surgical outcomes of strabismus surgery in TED. There had been no previous study showing significant changes in proptosis after strabismus surgery in TED. Furthermore, the main goal of strabismus surgery in TED is to obtain binocular single vision in the primary position, not exactly in the secondary and tertiary positions of gaze. Nonetheless, the study has also shown the results of postoperative field of binocular single vision, which indicates the presence of diplopia in any position of gaze.

In relation to the study, however, there can be changes in the lower eyelid position after inferior rectus muscle recession in TED. Since the paper had no data regarding this change, the authors, therefore, added the following sentence in page 11, lines 397-399 as a limitation: Previous studies have shown that IR muscle recession can result to lower eyelid retraction. There was no data on changes in the lower eyelid position after surgery, hence presenting as another limitation of the study. We are hoping that the findings presented in this study can still serve as a guide when managing TED-related strabismus.

  1. Ordinal Scale for Grading: The study mentions that the study used an ordinal scale of 0 to 3 for grading ocular motility restriction, deviating from the more commonly used 0 to 4 scale. This deviation from the standard scale could potentially introduce confusion and limit the comparability of the study's results with other research in the field. A justification for this choice is needed.

Reply: Thank you for pointing this out. The authors have acknowledged such limitation in page 11, lines 399-402. However, the use of this certain grading scale is based on the following range of ductions (0 = duction >45°, 1 = 30–45°, 2 = 15–30°, and 3 < 15°) (page 3, lines 103-104). Previous studies have also used the same grading scale (references #18 and 19) and there were no sources indicating or establishing a “standard scale” for grading ocular motility restriction. We do agree in the importance of having a standardized grading system, yet we hope that this study will still be acceptable for publication in order to provide further knowledge on this subject matter.

  1. Limited Generalizability: TED can manifest differently in patients, and factors such as disease severity and the presence of other associated eye conditions can vary widely. The study's findings may not be directly applicable to all TED patients, as the sample may not represent the full diversity of TED cases seen in clinical practice. 

Reply: Thank you for your insightful comment. As the reviewer has mentioned, ocular conditions do vary in TED. However, the present study showed no statistically significant difference in cross-sectional areas of the inferior and superior rectus muscles between successful and unsuccessful cases. Our previous study (reference #2) also showed no relationship between correction of excyclotropia and inferior rectus muscle thickness or fat degeneration in the inferior rectus muscle. The authors believe that the main factor affecting correction of strabismus in TED is the amount of muscle recession, resection, and transposition, not so much from other ocular conditions unrelated to the causality of strabismus, and that the results of the present study will be helpful for predicting the surgical outcomes of bilateral inferior rectus muscle recession in TED in clinical practice.

In summary, while the study provides valuable insights, these weaknesses, such as the small sample size, short follow-up period, and lack of detailed surgical methods, should be considered when interpreting the results and designing future research. Addressing these limitations could enhance the study's validity and applicability to a broader range of TED patients.

Reply: Thank you for your comment. We hope that the revised manuscript will be acceptable for publication in this esteemed journal.

Reviewer 3 Report (New Reviewer)

Comments and Suggestions for Authors

Dear Editor

The manuscript entitled “Surgical Outcomes of Bilateral Inferior Rectus Muscle Recession for Restrictive Strabismus Secondary to Thyroid Eye Disease” has been reviewed. The retrospective study reported 3 months of outcomes of bilateral IR recession in 12 TED-related strabismus. I think there are some critiques:  

Introduction

1.     I think, the “Introduction” could be shortened to make it much more succinct which would help highlight the key message

2.     The sentence “approximately 15% of patients with TED suffering from symptomatic ocular motility” is repeated in lines 31 and 43.

3.     Lines 94-98: there is no need to mention the sample size and variables in the Introduction.

Materials and Methods:

4.     Lines 117 and 118 “Restriction of upward gaze was graded on an ordinal scale 117 from 0 (normal) to 3 (severely restricted)….” What is the meaning of severely restricted?

5.     Lines 150 and 151: the sentence “The dose used for orbital radiotherapy was 20 Gy in all the treated patients.” Should transferred to the Results.

6.     The amount of IR recession in the eye with grade 1 was 2 mm, actually equal to the amount of resection effect of the secured sutures!!

Results:

7.     There were 3 patients with preoperative A-pattern that improved after IR recession without nasal transposition. What was the cause of the pattern in those cases? Were there concomitant incyclotorsion in those cases? How did the pattern resolve after IR recession without nasal transposition?

Author Response

Reviewer #3

Dear Editor

The manuscript entitled “Surgical Outcomes of Bilateral Inferior Rectus Muscle Recession for Restrictive Strabismus Secondary to Thyroid Eye Disease” has been reviewed. The retrospective study reported 3 months of outcomes of bilateral IR recession in 12 TED-related strabismus. I think there are some critiques: 

Introduction

  1. I think, the “Introduction” could be shortened to make it much more succinct which would help highlight the key message

Reply: Thank you for your suggestion. We have shortened the introduction section. We hope that this shortened version will be acceptable for the reviewer.

  1. The sentence “approximately 15% of patients with TED suffering from symptomatic ocular motility” is repeated in lines 31 and 43.

Reply: Thank you for pointing this out. We have deleted the corresponding sentence in response to comment #1 as well.

  1. Lines 94-98: there is no need to mention the sample size and variables in the Introduction.

Reply: Thank you for your suggestion. We have deleted the sample size and variables from the introduction section.

Materials and Methods:

  1. Lines 117 and 118 “Restriction of upward gaze was graded on an ordinal scale 117 from 0 (normal) to 3 (severely restricted)….” What is the meaning of severely restricted?

Reply: We have changed the phrase “from 0 (normal) to 3 (severely restricted)” to “(0 = duction >45°, 1 = 30–45°, 2 = 15–30°, and 3 < 15°)” in page 3, lines 103-104.

  1. Lines 150 and 151: the sentence “The dose used for orbital radiotherapy was 20 Gy in all the treated patients.” Should transferred to the Results.

Reply: Thank you for your suggestion. We understand the impression this statement presents, but the authors simply intended to present the dose of radiotherapy as additional information in relation to data collection of patient history with steroid pulse therapy and/or orbital radiotherapy, as mentioned in page 3, lines 137-138. We also presented the number of patients who underwent radiotherapy in Table 1 as a result of this study. We hope that this clarifies the reason for having the statement under data collection and we would highly appreciate your understanding.

  1. The amount of IR recession in the eye with grade 1 was 2 mm, actually equal to the amount of resection effect of the secured sutures!!

Reply: Thank you for this comment. The authors would like to clarify if this is pertaining to the resection effect on the contralateral eye, given that the patient underwent correction of non-restrictive strabismus of one eye. If so, this is not applicable to our study because the subjects have fibrotic extraocular muscles secondary to the inflammatory changes caused by thyroid orbitopathy. Hence, weakening the IR muscle on one eye does not produce a strengthening effect on the contralateral eye, necessitating the bilateral IR recession procedure to correct the deviation on primary gaze.

Results:

  1. There were 3 patients with preoperative A-pattern that improved after IR recession without nasal transposition. What was the cause of the pattern in those cases? Were there concomitant incyclotorsion in those cases? How did the pattern resolve after IR recession without nasal transposition?

Reply: Thank you for your questions. There were no sources that could explain resolution of A-pattern strabismus after IR recession without nasal transposition, but the authors are speculating that the restriction of the fibrotic IR muscle was causing compensatory over-action of the antagonist superior rectus muscle (resulting to incyclotorsion/adduction on elevation) and weakening the IR muscle eventually led to resolution of the SR over-action as well (we added this information in page 11, lines 373-378). Since TED-related strabismus has a different mechanism from pediatric or neurologic strabismus, this unusual phenomenon has yet to be investigated upon. As mentioned in page 11, lines 371-373, our previous study indicated that unilateral IR muscle recession ± IR nasal transposition does not increase the risk of pattern strabismus (reference #10). The same conclusion may be applicable to bilateral IR muscle recession. In addition, a previous study (reference #15) showed that a thicker IR muscle caused A-pattern strabismus after IR muscle recession in TED. We compared the thickness of the IR muscle between cases with and without A-pattern strabismus. Consequently, the IR muscle in more severe eyes was thinner in cases with A-pattern strabismus (P = 0.036). This may also affect improvement of A-pattern strabismus in our study. We added this content in page 11, lines 378-384.

None of the patients with A-pattern strabismus had incyclotropia before and after surgery. In the 3 cases with A-pattern strabismus, although the horizontal deviation angle in both 15° upward and downward gazes reduced after surgery, that in upward gaze more decreased, resulting in improvement of A-pattern strabismus. We added this information in page 8, lines 270-273.

Reviewer 4 Report (New Reviewer)

Comments and Suggestions for Authors

The authors describe the outcomes of 12 patients who underwent asymmetric bilateral IR recessions with nasalward transpostion for TED. They undertook a detailed analysis using regression techniques to determine the relationship between surgical dosage and outcome. However, several premises are problematic. First, the reason given for nasalward transpostion of the IRc was to reduce excyclotorsion. However, none of the study patients in the "successful outcome" group had 10 degrees or more of excylotorsion that would create a barrier to fusion, so the need for nasalward transposition did not exist.  Why was the transposition performed? Additionally, in the "more severe eyes" (I could not find a definition for use of this term on line 300), change in cyclotorsion was correlated with the amount of IR recession, so was there any reason to address cyclotorsion with nasalward transposition? Was the cyclotorsion not addressed with the IR recession alone? The conclusion of Kushner's study involving A pattern exotropia (lines 364-374) was difficult to understand. The authors state that they did not transpose inferior recti in 3 patients who had A pattern strabismus pre-operatively, but did transpose in patients who didn't have pattern strabismus and did not induce pattern strabismus in these patients (which is not something that Kushner or others cited have claimed or supported). But at the end of the paragraph, they justify transposition with citations stating that transposition is helpful for A pattern strabismus. It would appear that Dr. Kushner's warning about inducing incyclotropia by transposing tight IR in TED (Kushner BJ. Torsion and pattern strabismus: potential conflicts in treatment. JAMA Ophthalmology 2013;131:190-3.) held true for this study, as negative values of excyclotropia resulted  in the unsuccessful group (assuming that negative values of excycotropia equate with incyclotropia). The analyses seemed to substantiate a basic understanding of surgeons who care of TED patients with strabismus: the more severe cases tend not to follow surgical paradigms for dose-response and require additional surgery and cannot be corrected with "calculated" amounts of surgery.  The authors should conclude with a better description of how their results will inform their future surgical planning. And a final note, reporting in degrees of hypertropia and esotropia is problematic for many surgeons, as we standardly use prism diopters in clinical studies.

Author Response

Reviewer #4

The authors describe the outcomes of 12 patients who underwent asymmetric bilateral IR recessions with nasalward transpostion for TED. They undertook a detailed analysis using regression techniques to determine the relationship between surgical dosage and outcome. However, several premises are problematic. First, the reason given for nasalward transpostion of the IRc was to reduce excyclotorsion. However, none of the study patients in the "successful outcome" group had 10 degrees or more of excylotorsion that would create a barrier to fusion, so the need for nasalward transposition did not exist.  Why was the transposition performed? Additionally, in the "more severe eyes" (I could not find a definition for use of this term on line 300), change in cyclotorsion was correlated with the amount of IR recession, so was there any reason to address cyclotorsion with nasalward transposition? Was the cyclotorsion not addressed with the IR recession alone? The conclusion of Kushner's study involving A pattern exotropia (lines 364-374) was difficult to understand. The authors state that they did not transpose inferior recti in 3 patients who had A pattern strabismus pre-operatively, but did transpose in patients who didn't have pattern strabismus and did not induce pattern strabismus in these patients (which is not something that Kushner or others cited have claimed or supported). But at the end of the paragraph, they justify transposition with citations stating that transposition is helpful for A pattern strabismus. It would appear that Dr. Kushner's warning about inducing incyclotropia by transposing tight IR in TED (Kushner BJ. Torsion and pattern strabismus: potential conflicts in treatment. JAMA Ophthalmology 2013;131:190-3.) held true for this study, as negative values of excyclotropia resulted in the unsuccessful group (assuming that negative values of excycotropia equate with incyclotropia). The analyses seemed to substantiate a basic understanding of surgeons who care of TED patients with strabismus: the more severe cases tend not to follow surgical paradigms for dose-response and require additional surgery and cannot be corrected with "calculated" amounts of surgery.  The authors should conclude with a better description of how their results will inform their future surgical planning. And a final note, reporting in degrees of hypertropia and esotropia is problematic for many surgeons, as we standardly use prism diopters in clinical studies.

Reply: Thank you for your comment.

First, we performed nasal transposition of the IR muscle in patients who were aware of torsional diplopia before surgery and preoperative excyclotropia angle was larger than estimated excyclotropic angle correction after IR muscle calculated with the following formula: 0.4° of excyclotropic angle correction per 1mm IR muscle recession. We added this information in page 5, lines 200-202.

We defined more hypotropic eyes with more severe restriction of supraduction as more severe eyes. This was mentioned in page 5, lines 194-195.

As the reviewer has mentioned, the amount of nasal IR muscle transposition was not correlated with the correction of excyclotropia in the successful cases, this was correlated with the correction of excyclotropia in total (see, Table 4). We, therefore, believe that nasal IR muscle transposition was necessary to correct excyclotropia in our patients.

As mentioned in the response for comment #7 from reviewer #3, resolution of the A-pattern strabismus after surgery in the 3 cases with preoperative A-pattern strabismus has yet to be investigated upon. In contrast, patients who underwent nasal IR muscle transposition did not show A-pattern strabismus after surgery, implying that nasal IR muscle transposition may prevent postoperative A-pattern strabismus in these cases. As the reviewer indicated that although excyclotropia was overcorrected (4 degrees incyclotropia) in 1 unsuccessful case, excyclotropia was adequately corrected in the other 11 unsuccessful cases (we added this information in page 8, lines 282-284, and Tables 2 and 3). We, therefore, believe that targeting angles of vertical and torsional deviations with remaining 2-3° hypotropia and 1-2°excyclotropia (mentioned in page 5, lines 207-208, and page 11, lines 385-386) is acceptable for bilateral IR muscle recession.

Indeed, the use of prism diopters is better for clinical studies on strabismus. However, since measuring deviation with prisms involves estimation by an examiner, measurement can become subjective. In this study, the use of a synoptophore provides a more objective measurement of the deviation angles, and the deviation angles are displayed at the unit of degrees. We, therefore, incorporated the unit of degrees for consistency of measurement in this study. We hope that this will be acceptable to minimize potential bias for more accurate findings.

Round 2

Reviewer 3 Report (New Reviewer)

Comments and Suggestions for Authors

Dear Editor

The revised manuscript entitled “Surgical Outcomes of Bilateral Inferior Rectus Muscle Recession for Restrictive Strabismus Secondary to Thyroid Eye Disease” has been reviewed. I appreciate the authors for their efforts. The issues have been addressed in the revised manuscript. The manuscript is acceptable in its present form.

Kind regards

This manuscript is a resubmission of an earlier submission. The following is a list of the peer review reports and author responses from that submission.

Round 1

Reviewer 1 Report

Comments and Suggestions for Authors

It is an interesting subject and well-written paper about the “Surgical Outcomes of Bilateral Inferior Rectus Muscle Recession for Restrictive Strabismus Secondary to Thyroid Eye Disease”. The author should be explained more clearly about the result of this research and its clinical applications in the final discussion and conclusion parts.

Comments on the Quality of English Language

Acceptable

Author Response

Point-by-point Response to Reviewers’ Comment

We wish to thank the reviewers for taking the time to help us improve our manuscript. We value the comments and we hope that the changes we made will now make our manuscript acceptable to the reviewers.

In addition, we found additional 2 patients with thyroid eye disease who underwent asymmetric bilateral inferior rectus recession. We added them in this study and reanalyzed the results.

Reviewer #1

It is an interesting subject and well-written paper about the “Surgical Outcomes of Bilateral Inferior Rectus Muscle Recession for Restrictive Strabismus Secondary to Thyroid Eye Disease”. The author should be explained more clearly about the result of this research and its clinical applications in the final discussion and conclusion parts.

Thank you for this comment! We have revised the conclusion to convey a better picture of the impact and clinical application of our study. We hope that the revision is sufficient to address this concern.

Reviewer 2 Report

Comments and Suggestions for Authors

This retrospective study examined the outcomes of bilateral inferior rectus (IR) muscle recession in thyroid eye disease. Ten patients with restrictive vertical strabismus were included. Surgical success was determined based on specific criteria, and the study found that 7 out of 10 patients achieved successful outcomes. The results showed a correlation between changes in deviation angles and the amount of muscle recession and nasal transposition. These findings have implications for determining optimal surgical approaches in thyroid-related bilateral IR myopathy.

Introduction

The introduction lacks essential background information on thyroid eye disease (TED), such as its prevalence, symptoms, and impact on patients.

The introduction does not sufficiently review previous studies on surgical outcomes in TED.

The introduction does not clearly state the rationale or justification for conducting the study.

The specific objectives of the study are not clearly stated.

The introduction does not provide adequate information about the study population, including the selection criteria or demographic characteristics.

The introduction briefly mentions the difficulties in achieving binocular single vision in TED patients but does not elaborate on the specific challenges faced during strabismus surgery or the reasons for high reoperation rates.

The introduction mentions a lack of published reports on bilateral inferior rectus (IR) recession in TED but does not discuss or analyze the limited existing literature.

The introduction does not present any hypotheses or specific research questions that the study aims to address.

Method

The section on ethics approval lacks details on the specific ethical considerations addressed by the institutional review board (IRB) of Aichi Medical University Hospital.

The rationale for obtaining a waiver of informed consent is not thoroughly explained. While the retrospective nature of the study may justify the waiver

It is essential to clearly outline the study design, including the rationale for choosing a retrospective, observational approach.

The description of the diagnosis of thyroid eye disease (TED) is brief and lacks clarity.

The section on data collection lacks specific information on how the various data points were collected.

The section on statistical analyses is concise and lacks sufficient details.

The section does not mention how the results of the statistical analyses were reported or interpreted.

Author Response

Point-by-point Response to Reviewers’ Comment

We wish to thank the reviewers for taking the time to help us improve our manuscript. We value the comments and we hope that the changes we made will now make our manuscript acceptable to the reviewers.

In addition, we found additional 2 patients with thyroid eye disease who underwent asymmetric bilateral inferior rectus recession. We added them in this study and reanalyzed the results.

Reviewer #2

This retrospective study examined the outcomes of bilateral inferior rectus (IR) muscle recession in thyroid eye disease. Ten patients with restrictive vertical strabismus were included. Surgical success was determined based on specific criteria, and the study found that 7 out of 10 patients achieved successful outcomes. The results showed a correlation between changes in deviation angles and the amount of muscle recession and nasal transposition. These findings have implications for determining optimal surgical approaches in thyroid-related bilateral IR myopathy.

Introduction

The introduction lacks essential background information on thyroid eye disease (TED), such as its prevalence, symptoms, and impact on patients.

Thank you for your comment. The authors initially decided to focus on information that is most relevant to the study, which is the effect on extraocular muscles. We have also mentioned the impact on patients, i.e. difficulty in performing activities of daily living. Most studies on TED found in literature have already emphasized prevalence and symptomatology. Nonetheless, we have revised the introduction to include such information (page 1, lines 32-39) and also for the sake of the readers who are not yet familiar with the implications of TED.

The introduction does not sufficiently review previous studies on surgical outcomes in TED.

Thank you for pointing this out. The authors have included some relevant studies on bilateral inferior rectus muscle recession and the associated outcomes (page 2, lines 55-57, 59-65, and 67-68). We hope that the revision is sufficient.

The introduction does not clearly state the rationale or justification for conducting the study.

Thank you for this comment. Firstly, although there have been 5 reports presenting the results of bilateral inferior rectus muscle recession in TED, 4 out of the 5 reports included only 3-8 patients. In addition to this, 2 out of the 5 reports included patients who underwent prior orbital decompression, which likely affects the surgical outcomes of strabismus surgery. Furthermore, only one report presented the dose-effect relationship between improved ocular elevation and the amount of inferior rectus muscle recession (reference #11), but this study did not determine the relationship between the changes in ocular deviation angles and the amount of inferior rectus muscle recession with or without nasal transposition. We, therefore, examined the surgical outcomes of asymmetric bilateral inferior rectus muscle recession in TED and analyzed the relationship between changes in ocular deviation angles and amount of inferior rectus muscle recession and nasal transposition. The authors have revised the introduction and have mentioned these statements in page 2, lines 67-77. We hope that the study is relevant for publication in order to provide more evidence for effectively managing TED patients with restrictive strabismus.

The specific objectives of the study are not clearly stated.

Thank you for pointing this. We have included the specific objectives of the study at the last portion of the introduction (page 2, lines 75-81). We hope that this clarifies the objectives of the present study.

The introduction does not provide adequate information about the study population, including the selection criteria or demographic characteristics.

Thank you for this comment. The authors have mentioned in the methods section under “Study Design” the specific study population and selection criteria, which was patients diagnosed with TED at our institution and who underwent asymmetric bilateral IR recession with or with our nasal transposition by the corresponding author. The demographic characteristics were determined retrospectively, as mentioned under “Data Collection,” since the study was not prospective in nature. Hence, the authors did not mention such information in the introduction. We hope that this clarifies the lack of information in the first part of the paper.

The introduction briefly mentions the difficulties in achieving binocular single vision in TED patients but does not elaborate on the specific challenges faced during strabismus surgery or the reasons for high reoperation rates.

Thank you for this comment. The authors have revised the introduction in order to include the reason for the challenges faced during strabismus surgery and the main reason for high reoperation rates, i.e. undercorrection, overcorrection, and postoperative torsional deviations (page 2, lines 47-48). We hope that the revision will suffice.

The introduction mentions a lack of published reports on bilateral inferior rectus (IR) recession in TED but does not discuss or analyze the limited existing literature.

Thank you for pointing this out. As mentioned in the reply for comment #3, we have revised the introduction to include a short discussion on the previous studies on bilateral IR recession. We hope that this sufficiently addresses the comment.

The introduction does not present any hypotheses or specific research questions that the study aims to address.

Thank you for this comment. As mentioned in the reply for comment #4, we have revised the introduction to include the main research question, which is to determine the effectiveness of asymmetric bilateral IR recession for the management of restrictive strabismus in TED patients.

Method

The section on ethics approval lacks details on the specific ethical considerations addressed by the institutional review board (IRB) of Aichi Medical University Hospital.

Thank you for this comment. The IRB of our institution has decided on the statement to be used on this research paper conducted in Aichi Medical University Hospital that will be submitted for publication, so the authors have simply complied to such statement written under ethics approval. The same statement has been used in previous observational studies that were approved for publication. Since the authors cannot change the statement provided by the IRB, we hope that this explanation will be sufficient to address the concern.

The rationale for obtaining a waiver of informed consent is not thoroughly explained. While the retrospective nature of the study may justify the waiver

Thank you for pointing this out. The Japanese Ministry of Health has declared that observational studies, whether prospective or retrospective in nature, do not require informed consent. Hence, the IRB of our institution has granted a waiver, which was stated in the methods section under ethics approval. We hope that this sufficiently addresses the concern.

It is essential to clearly outline the study design, including the rationale for choosing a retrospective, observational approach.

Thank you for this comment. We have included the statement in the methods section regarding the rationale of choosing a retrospective, observational study design (page 3, lines 105-108).

The description of the diagnosis of thyroid eye disease (TED) is brief and lacks clarity.

Thank you for pointing this out. The authors have revised the statement under “Diagnosis of TED,” which is simply the presence of eyelid signs alongside existing autoimmune thyroid disorder that was diagnosed by a previous clinician. Please refer to the methods section (page 3, lines 111-117).

The section on data collection lacks specific information on how the various data points were collected.

Thank you for this comment. We have included a statement regarding the method of data collection through our electronic medical records. Since the study is a retrospective chart review, the data were collected from the electronic chart of every patient (page 3, lines 123-125). We hope that this clarifies the data collection process.

The section on statistical analyses is concise and lacks sufficient details.

Thank you for pointing out. We included additional information in the “Statistical Analyses” section (page 6, lines 217-219 and lines 226-227). We hope the additional information will suffice.

The section does not mention how the results of the statistical analyses were reported or interpreted.

Thank you for this comment. We have mentioned our interpretation of the results of statistical analyses in page 10, stated in the 2nd to 4th paragraphs. Nonetheless, the authors have added statements regarding interpretations of the results in page 10, lines 326-328, lines 342-348, and lines 351-352. We hope that this addresses the concern.

Round 2

Reviewer 2 Report

Comments and Suggestions for Authors

Despite the initial round of revisions, it appears that the authors have not adequately addressed several critical points. The introduction still lacks essential background information on thyroid eye disease (TED), including its prevalence, symptoms, and impact on patients. Moreover, previous studies on surgical outcomes in TED have not been sufficiently reviewed. The introduction does not clearly state the rationale for conducting the study or the specific objectives. Furthermore, important details about the study population, such as selection criteria and demographic characteristics, are still lacking. The challenges faced during strabismus surgery in TED patients and the reasons for high reoperation rates have not been adequately addressed. The limited existing literature on bilateral inferior rectus (IR) recession in TED has not been discussed or analyzed. Finally, the introduction does not present any hypotheses or specific research questions that the study aims to address. These unresolved issues undermine the clarity and significance of the study's findings.